# Small intestinal microbial dysbiosis underlies symptoms associated with functional gastrointestinal disorders

George B. Saffouri[1,14], Robin R. Shields-Cutler[2,3,14], Jun Chen[4], Yi Yang[5], Heather R. Lekatz[1], Vanessa L. Hale[6], Janice M. Cho[7], Eric J. Battaglioli[1], Yogesh Bhattarai[1], Kevin J. Thompson[4], Krishna K. Kalari[4], Gaurav Behera[1], Jonathan C. Berry[8], Stephanie A. Peters[1], Robin Patel[8], Audrey N. Schuetz[8], Jeremiah J. Faith[9], Michael Camilleri[1,10], Justin L. Sonnenburg[11], Gianrico Farrugia[12], Jonathan R. Swann [iD] [5], Madhusudan Grover [iD] [1,10], Dan Knights[2,13] & Purna C. Kashyap [iD] [1,10]

Small intestinal bacterial overgrowth (SIBO) has been implicated in symptoms associated with functional gastrointestinal disorders (FGIDs), though mechanisms remain poorly defined and treatment involves non-specific antibiotics. Here we show that SIBO based on duodenal aspirate culture reflects an overgrowth of anaerobes, does not correspond with patient symptoms, and may be a result of dietary preferences. Small intestinal microbial composition, on the other hand, is significantly altered in symptomatic patients and does not correspond with aspirate culture results. In a pilot interventional study we found that switching from a high fiber diet to a low fiber, high simple sugar diet triggered FGID-related symptoms and decreased small intestinal microbial diversity while increasing small intestinal permeability. Our findings demonstrate that characterizing small intestinal microbiomes in patients with gastrointestinal symptoms may allow a more targeted antibacterial or a diet-based approach to treatment.

[1] Division of Gastroenterology and Hepatology, Mayo Clinic, Rochester, MN 55902, USA. [2] BioTechnology Institute, College of Biological Sciences, University of Minnesota, Minneapolis, MN 55455, USA. [3] Department of Biology, Macalester College, Saint Paul, MN 55105, USA. [4] Division of Biomedical Statistics and Informatics, Department of Health Sciences Research, Mayo Clinic, Rochester, MN 55902, USA. [5] Computational and Systems Medicine Section of the Department of Surgery and Cancer, Imperial College, (London), UK. [6] Department of Veterinary Preventive Medicine, The Ohio State University, Columbus, OH 43210, USA. [7] Division of Internal Medicine, Mayo Clinic, Rochester, MN 55902, USA. [8] Division of Clinical Microbiology, Department of Laboratory Medicine and Pathology, Mayo Clinic, Rochester, MN 55902, USA. [9] Department of Genetics and Genomic Sciences, Medicine, and Clinical Immunology, Icahn School of Medicine at Mount Sinai, New York, NY 10029, USA. [10] Department of Physiology and Biomedical Engineering, Mayo Clinic, Rochester, MN 55902, USA. [11] Department of Microbiology and Immunology, Stanford University, Stanford, CA 94305, USA. [12] Division of Gastroenterology, Mayo Clinic, Jacksonville, FL 32224, USA. [13] Department of Computer Science and Engineering, University of Minnesota, Minneapolis, MN 55455, USA. [14] These authors contributed equally: George B. Saffouri and Robin R. Shields-Cutler. Correspondence and requests for materials should be addressed to P.C.K. (email: kashyap.purna@mayo.edu)

The human gut microbiome has emerged as an important factor in the pathogenesis of functional gastrointestinal (GI) disorders (FGIDs). Its role in modulating important physiological processes such as GI motility and secretion, maintenance of epithelial barrier integrity, and its role in communication between the gut and the central nervous system may underlie its contribution to symptoms associated with FGIDs such as irritable bowel syndrome (IBS)[1,2]. The majority of microbiome studies focus on stool, which is broadly reflective of the GI microbiome, and the colonic mucosa, where microbes may play a role in triggering inflammation[3]. The small intestinal microbiome, however, remains relatively unexplored in most GI disorders including FGIDs. This is relevant, as a quantitative increase in the small intestinal bacteria, characterized as small intestinal bacterial overgrowth (SIBO), has been implicated in symptoms associated with FGIDs, though mechanisms underlying this effect remain poorly defined.

Few studies have attempted to characterize the small intestinal microbiota, primarily owing to the difficulty in accessing the small intestine and lower microbial density that makes it difficult to obtain sufficient bacterial DNA[4]. Recent studies comparing small bowel mucosal microbiota (obtained via either Watson capsule biopsy or enteroscopy with biopsy) in IBS patients and healthy volunteers in Sweden and in Taiwan found either no changes or IBS-associated increases in *Prevotella spp.*, respectively[5,6]. There are additional small studies that have characterized the small intestinal microbial composition using ileostomy effluent samples that allow for easier collection[7,8], but are not necessarily reflective of normal small intestinal microbiota. The ileum primarily serves as a reservoir[9] and exhibits significant differences in motor function when compared with the upper small intestine, which acts as a conduit[10] with little stasis and lower bacterial counts. The bolus transport in the duodenum, however, may be decreased or associated with retrograde bolus transfer events in diabetic gastroparesis[11].

SIBO is implicated in driving GI symptoms such as diarrhea, abdominal pain, and bloating; the gold standard test for SIBO is a duodenal aspirate culture to measure microbial density in the duodenum[12-14]. SIBO is currently defined as $\geq 10^5$ CFU/mL (colony-forming units per ml) on aspirate culture, based on the initial description in Billroth II anatomy patients with stagnant loop syndrome[15-17]. This has been extrapolated for the diagnosis of SIBO in other conditions. This approach to diagnosis is limited by the inability to differentiate disease-promoting microbes from potentially beneficial microbes and the inability to sample microbes in different regions of the small intestine. There is wide variability in antibiotic treatment response among patients diagnosed with SIBO, a potential worsening of GI symptoms[18] with antibiotics, and though some data do exist[19], there are few robust randomized controlled trials assessing the efficacy of specific antibiotics for SIBO[20].

Recent advances in next-generation sequencing have significantly improved our ability to characterize complex microbial communities and determine changes in microbiota composition and function that may contribute to disease states such as IBS where microbiota changes are associated with symptoms[21]. Diet is a key driver of gut microbial composition[22] and is also thought to have a role in FGIDs like IBS. A high-fiber diet has been associated with improvement in symptoms in IBS, based on systematic review and meta-analyses[23]. However, dietary influence, such as with fiber supplementation, on the small intestinal microbiome has not been studied.

In this work, we characterized the small intestinal microbiota of patients with GI symptoms undergoing testing for SIBO. We found significant alteration in the small intestinal microbial composition especially in a subset of symptomatic patients. The alteration indicated potential consequences for the functional capacity of the small intestinal microbiome specifically with regard to processing of dietary carbohydrate and fiber. SIBO, however, did not correlate with patient symptoms and was also seen in healthy individuals consuming a high-fiber diet. In a pilot intervention study performed subsequently in healthy individuals consuming a high fiber diet, we found a short-term switch to low-fiber diet leads to change in intestinal permeability and appearance of GI symptoms associated with a change in microbial diversity. Our findings suggest that characterizing the small intestinal microbial composition is important as it may allow a more-targeted antibiotic approach in symptomatic patients. At the same time, relying on quantitative culture of small intestinal aspirates alone may not be sufficient as these results may be influenced by diet.

## Results

**Duodenal aspirate cultures do not correlate with patient symptoms.** We evaluated 126 symptomatic (21% male; age 15–89 years, median 55 years) patients who underwent esophagogastroduodenoscopy (EGD) with duodenal aspirate collection; the major symptoms that led to investigation of small bowel bacterial counts included diarrhea (45%), abdominal pain (28%), and bloating (13%). A concomitant organic diagnosis was only noted in a minority of patients and included celiac disease ($n = 14$), microscopic colitis ($n = 6$), ulcerative colitis ($n = 5$), and pancreatic insufficiency ($n = 5$). In total, 27 (21.4%) patients underwent some form of GI surgery. These surgeries included Roux-en-Y gastric bypass ($n = 4$), ileocolonic valve resection ($n = 5$), small or large bowel resection ($n = 11$), pancreatectomy ($n = 3$), or partial or full gastrectomy ($n = 2$). The surgeries were not mutually exclusive.

Of the 126 patients, 66 (52%) tested positive for SIBO, whereas 60 patients (48%) tested negative for SIBO. Among the 66 who tested positive, 49 were positive for anaerobic bacterial overgrowth and 17 were positive for mixed anaerobic and aerobic overgrowth. None were positive for aerobic bacterial overgrowth alone. All healthy individuals tested negative for SIBO.

Interestingly, there was a positive correlation between SIBO and recent antibiotic exposure (odds ratio of 4.2; 95% CI: 1.05–24.3; $p = 0.028$, Fisher's exact test). There was no correlation between SIBO and any indication for testing ($p = 0.19$, Fisher's exact test with Monte Carlo simulation, 10,000 replicates), age, GI surgery, or proton-pump inhibitor (PPI) use (all $p > 0.1$, Fisher's exact test).

**Duodenal aspirate microbiome is altered in symptomatic patients.** We determined the 16 S rRNA-based microbial community composition in duodenal aspirates from the patients with GI symptoms ($n = 126$; Supplementary Data 1) and compared them with the microbial compositions of similarly collected duodenal aspirates from healthy volunteers ($n = 38$; 30% male; age 19–60 years, median 43 years).

Significant differences were observed in phylogenetic unweighted UniFrac-based ($p < 0.001$, permutational multivariate analysis of variance (PERMANOVA); Fig. 1a) and non-phylogenetic Bray–Curtis-based ($p < 0.001$, PERMANOVA; Fig. 1b) beta diversity between small intestinal microbial communities from symptomatic patients and healthy volunteers. There were significant differences in relative abundance at multiple taxonomic levels ($q < 0.05$, permutation test with $t$-statistic; Supplementary Fig. 1, Supplementary Data 2) when comparing the two communities, which included significant decreases in *Porphyromonas*, *Prevotella*, and *Fusobacterium* in symptomatic patients. The small intestinal microbial communities from symptomatic patients were characterized by

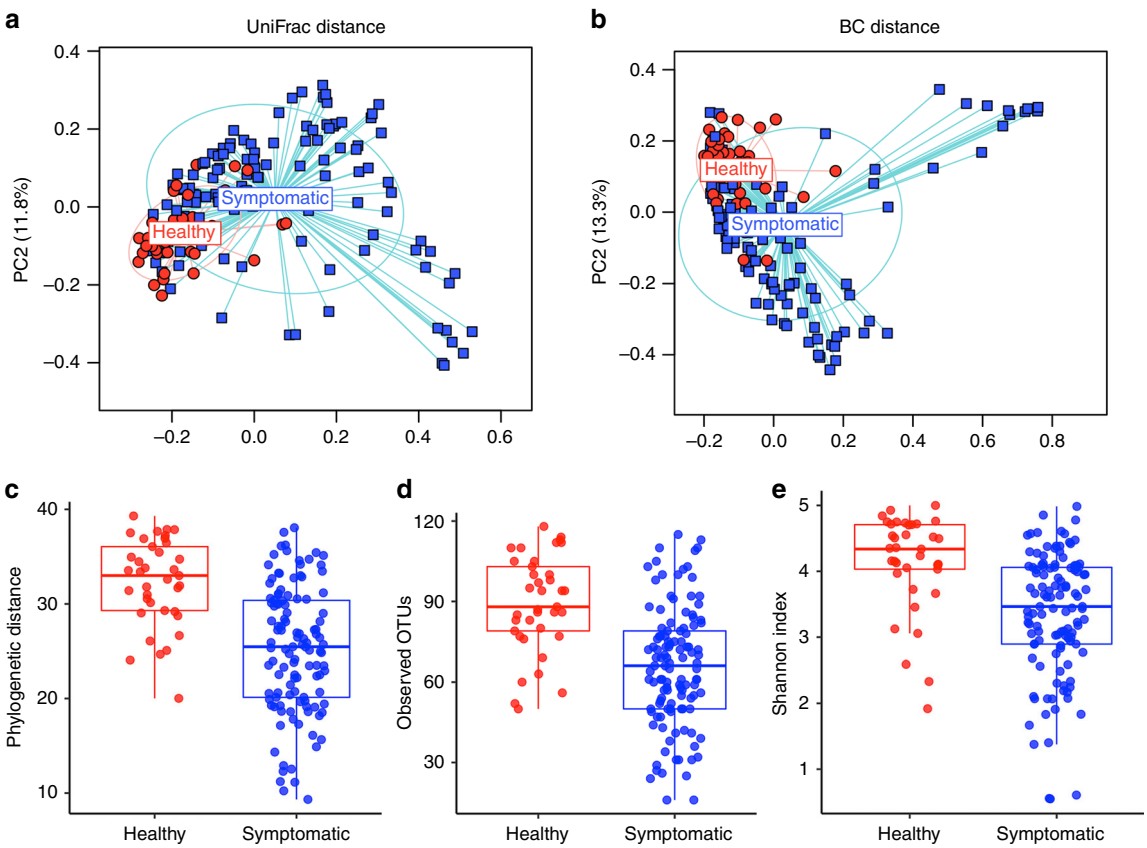

**Fig. 1** The duodenal microbiome is altered in patients with GI symptoms. Principal coordinate axis (PCoA) plot showing beta diversity of patients with GI symptoms ($n = 126$, blue) and healthy controls ($n = 38$, red) based on **a** unweighted UniFrac and **b** Bray–Curtis distances ($p = 0.001$, PERMANOVA). Alpha diversity (within subject) of patients with GI symptoms ($n = 126$, blue) and healthy controls ($n = 38$, red) based on **c** phylogenetic distance **d** observed OTUs and **e** Shannon diversity index metrics (rarefied to 5000 sequences; $p < 0.0001$, $t$ test). Tukey boxplots show the median with IQR and 1.5 IQR whiskers

significantly lower phylogenetic alpha diversity, richness, and evenness ($p < 0.0001$ for each, $t$- test; Fig. 1c–e).

Next, the primary microbial determinants responsible for the difference in small intestinal microbial composition in symptomatic patients were identified. We used Random Forest classification on the operational taxonomic unit (OTU)-level abundances to develop a symptom index (SI) model for microbial differences associated with symptomatic patients. The resulting index is the out-of-bag (OOB) predicted probability of symptomatic patient group membership; i.e., on a scale of 0 to 1, scores approaching 1 indicate high probability of a microbial community associated with GI symptoms. The SI differentiates symptomatic patients from healthy individuals (Fig. 2a), supported by receiver operating characteristic curve analysis (area under the curve = 0.896, $p < 0.0001$; 95% C.I.: 0.844–0.949, DeLong). Boruta selection yields 26 OTUs that significantly contribute to the classification (Fig. 2b). To determine whether patient characteristics contribute to changes in microbial composition we examined the clinical metadata obtained from patient medical health records. Advanced age, antibiotic use, history of GI surgery, and PPI use were found to be significantly associated with the SI (false discovery rate (FDR) $q < 0.1$, linear regression) and these four clinical factors together explain ~12.7% of variance ($R^2$) in the SI across all samples (Fig. 2c–f), whereas sex, body mass index (BMI), and SIBO were not significantly associated with SI.

**A subset of symptomatic patients have small intestinal microbial dysbiosis**. We found notable heterogeneity in the small intestinal microbial composition of symptomatic patients and a subset of patients appear compositionally distinct from healthy individuals in beta diversity plots (Fig. 1a, b). To further explore this observation, we re-classified samples from symptomatic patients as being healthy-like or dysbiotic based on their Aitchison distance to healthy small intestinal microbial communities using the CLOUD neighborhood method[24]. This classification scheme identified 37/38 (97%) healthy volunteers as "healthy-like", whereas 89 (71%) symptomatic patient communities were found to be healthy-like and 37 (29%) were found to be dysbiotic (based on distance from healthy, Fig. 3a). These groupings were comparable to those achieved by spectral clustering (Supplementary Table 1). The log-transformed CLOUD statistic for each sample defines the sample's "dysbiosis index" (DI) score (Fig. 3b). Boruta feature selection identified 23 genus-level taxa that contribute significantly to the DI, mostly by their relative absence from dysbiotic communities (Fig. 3c). We also found a significantly higher DI score in patients > 50 years old and those with history of antibiotic use, PPI use, or GI surgery ($p < 0.05$, $t$- test; Supplementary Fig. 2A–D), consistent with factors that contributed to the SI. In addition, there was a significant negative correlation between DI score and phylogenetic diversity, richness, and evenness ($p < 0.0001$, Pearson correlation; Fig. 3d–f).

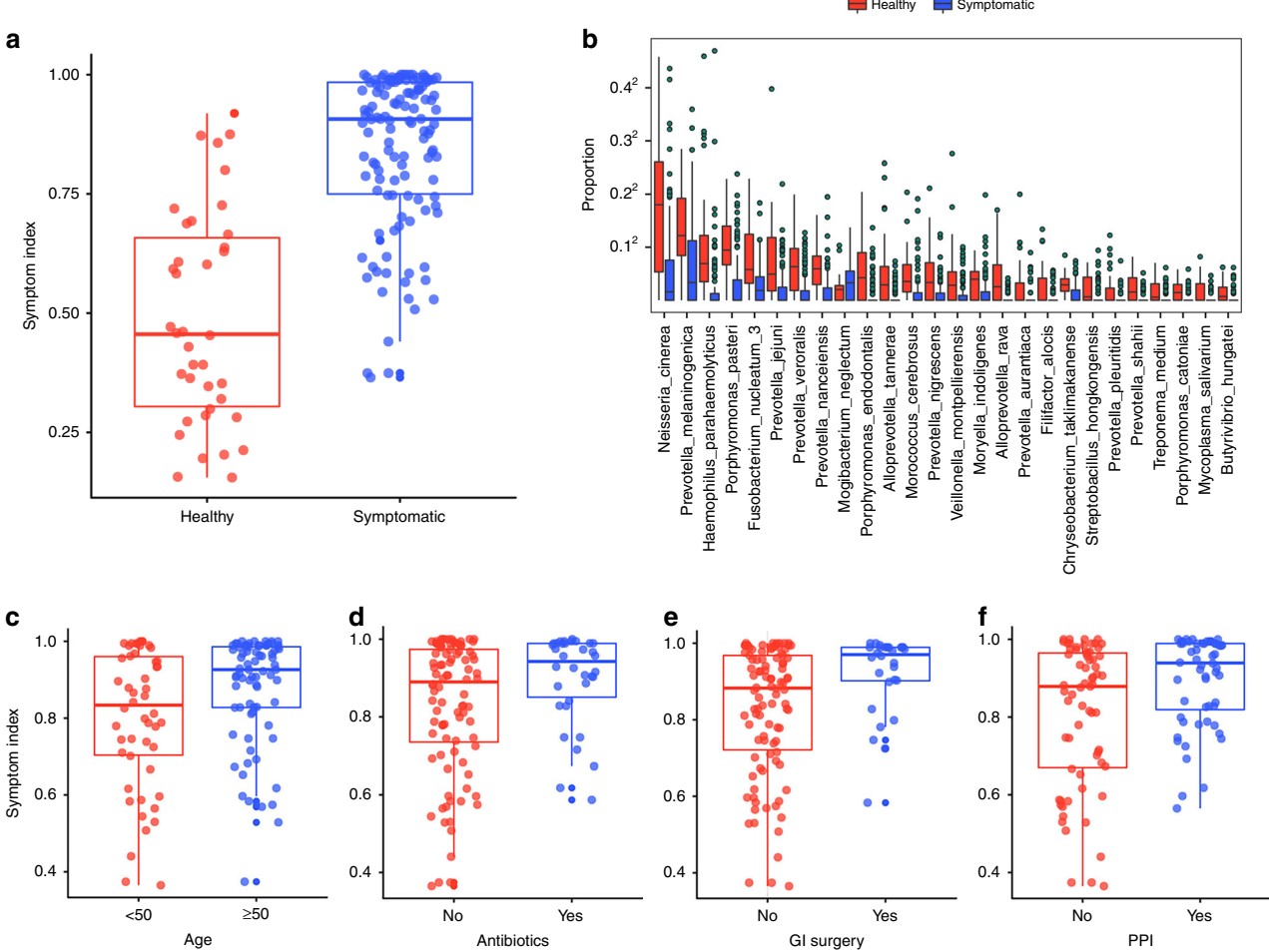

**Fig. 2** Symptom index differentiates healthy controls from patients with GI symptoms. **a** Symptom index indicating the probability (0–1) of being classified in the symptomatic patient group determined using Random Forests classification based on the individuals' OTU profiles. **b** Tukey boxplots show differences in relative abundance of the 26 OTUs among symptomatic patients ($n = 126$, blue) and healthy controls ($n = 38$, red) that significantly contribute to the Random Forest classification performance based on Boruta feature selection (boxplots show the median with IQR and 1.5 IQR whiskers). The four factors contributing to the variance in the symptom index among healthy controls (red) and symptomatic patients (blue) are **c** age **d** history of antibiotic use within 3 weeks **e** history of GI surgery and **f** PPI

**SIBO does not correlate with small intestinal microbial dysbiosis**. We compared the microbial composition of symptomatic patients with and without SIBO and found that SIBO does not correlate with small intestinal dysbiosis ($p = 0.33$, Fisher's exact test of culture result vs. dysbiosis classification; Fig. 4). This suggests that some symptomatic patients diagnosed with SIBO in fact have an overabundance of bacteria normally found in healthy microbial communities, whereas others who do not have SIBO based on quantitative assessment have dysbiosis as defined here. There are no significant differences in alpha or beta diversity among symptomatic patients with and without reported SIBO (Supplementary Fig. 3A–D).

**Imputed microbial functional pathways suggest differences in diet**. To determine the functional changes associated with differences in microbial composition, gut microbial function was imputed from the 16 S rRNA-based microbial composition using PICRUSt and pathway analysis. Differential predicted genes/pathways were determined using LEfSe (Supplementary Fig. 4A, B). Pathways reflective of oxidative stress such as ascorbate and aldarate metabolism and biosynthesis of siderophores were enriched in symptomatic patients, consistent with previous reports in dysbiosis. Interestingly, pathways associated with

simple sugar metabolism were enriched in symptomatic patients, whereas complex carbohydrate degradation pathways were more prevalent in healthy individuals. Patients' dietary histories were not available as these samples were obtained directly from the clinical microbiology laboratory but the increased prevalence of *Prevotella* and the enrichment of complex carbohydrate degradation pathways are suggestive of a higher fiber intake in healthy individuals while simple sugar metabolism pathways found in symptomatic patients may reflect a higher dietary intake of simple sugars. These data alone do not provide sufficient evidence to support the role of diet-related changes in small intestinal microbiome in causing GI symptoms, but do support this hypothesis. To better address this, a pilot dietary intervention study was performed.

**A subset of healthy individuals eating high-fiber diets have SIBO**. As GI symptoms were associated with decreased prevalence of *Prevotella* and potentially increased consumption of simple sugars, we next tested if a dietary change from a high-fiber diet to a high simple-sugar diet can trigger symptoms in a microbiota-dependent manner. Healthy individuals consuming baseline high fiber diets ( > 11 g/1000 cal; Supplementary Table 2) were identified and duodenal aspirates were obtained for

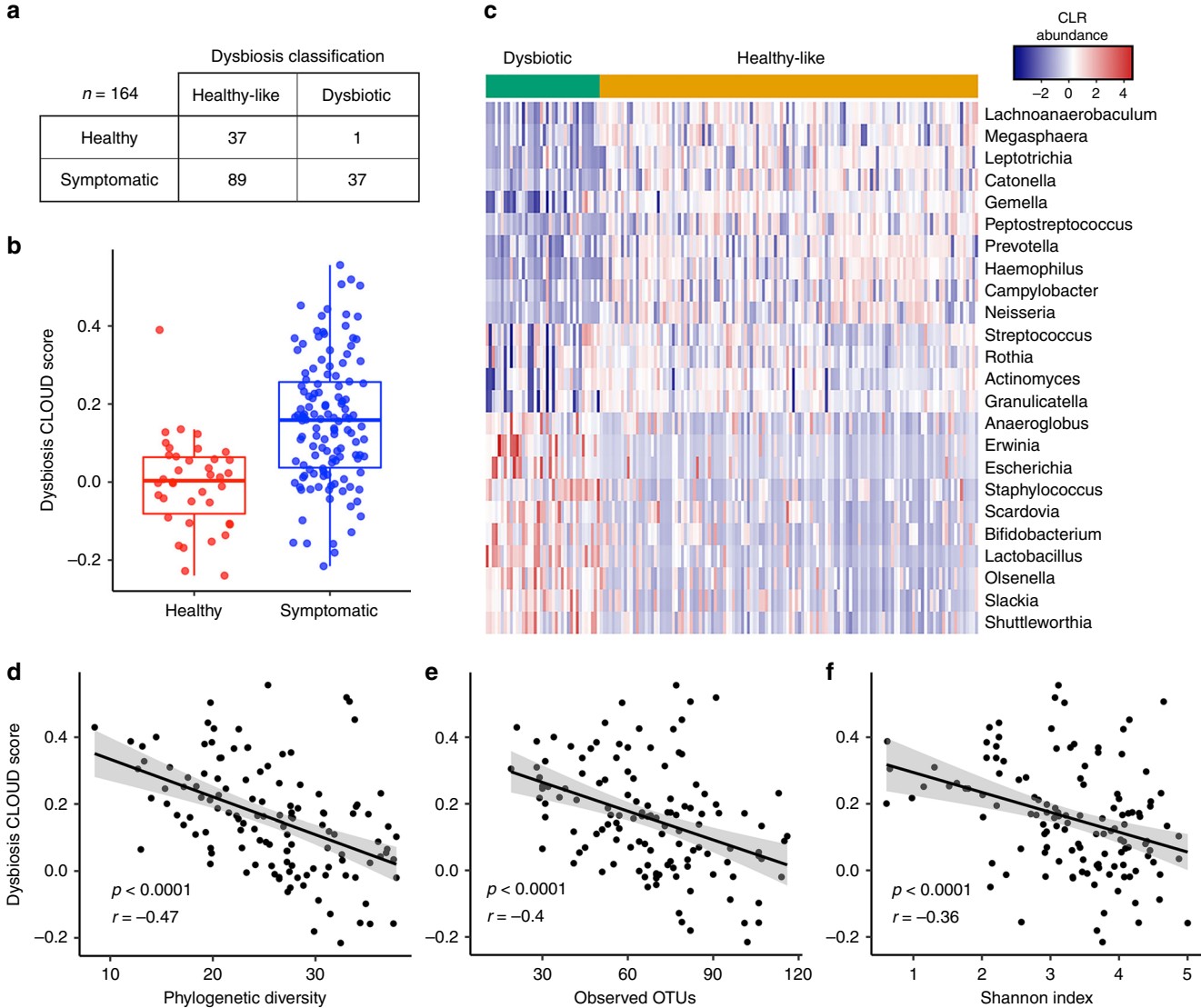

**Fig. 3** Dysbiosis index identifies a subset of symptomatic patients with altered microbial communities. **a** Classification of healthy and symptomatic patients as dysbiotic or healthy-like based on the CLOUD test, which evaluates each sample by its distance to healthy controls' microbiome distribution. **b** The log-transformed CLOUD statistic for samples (or dysbiosis index (DI)) from healthy controls ($n = 38$, red) and symptomatic patients ($n = 126$, blue). Tukey boxplots show the median with IQR and 1.5 IQR whiskers. **c** Heatmap of significantly different genus-level taxa from healthy control and symptomatic patients that contribute significantly to the DI identified using Boruta feature selection from the random forest model[40]. Samples classified as dysbiotic or healthy-like are indicated at the top by orange and green, respectively. Pearson correlation of DI with alpha diversity based on **d** phylogenetic diversity ($p < 0.0001$, $r = -0.47$, Pearson correlation), **e** observed OTUs ($p < 0.0001$, $r = -0.4$, Pearson correlation), **f** Shannon index ($p < 0.0001$, $r = -0.36$, Pearson correlation)

quantitative culture and small intestinal microbial community profiling using 16 S rRNA gene sequencing. Despite being asymptomatic, 8/16 (50%) subjects on a baseline high-fiber diet tested positive for SIBO by the standard culture criteria described above. Microbial community profiles were obtained from only 15 of the 16 participants after quality control and filtering of sequencing data. All subjects had a microbial community composition representative of a "healthy-like" community (Fig. 5a). The small bowel microbial communities of these high fiber-consuming healthy individuals clustered with the healthy individuals previously tested (Fig. 5b) based on the Aitchison beta diversity for each sample regardless of presence or absence of SIBO. The symptomatic patient microbiomes show wider distribution as noted previously. Accompanying boxplots show the distribution of principal coordinate 1, which accounts for 46% of the variance in data, further supporting the conclusion that

duodenal microbiome significantly distinguishes healthy and symptomatic individuals ($q < 0.0001$, pairwise $t$ test with FDR correction; Fig. 5b), but does not distinguish presence or absence of SIBO among healthy or symptomatic individuals ($q > 0.25$, pairwise $t$- test with FDR correction; Fig. 5b). This suggests that healthy individuals can have SIBO without any symptoms or alterations in microbial composition. There are no significant differences in small intestinal microbial alpha or beta diversity or microbial taxa among the healthy subjects with and without SIBO. Therefore, SIBO as currently defined may also result from dietary preferences, such as high fiber consumption, as shown here.

**Short-term diet change alters microbial diversity and triggers GI symptoms**. We then addressed whether diet-related changes

in small intestinal microbiota composition and function might be responsible, in part, for alterations in epithelial barrier function, and symptoms often associated with FGIDs. To investigate this, a pilot short-term dietary intervention study was performed limiting the consumption of fiber in the 16 healthy individuals consuming baseline high fiber diet ( > 11 g/1000 cal) identified above. All subjects were placed on a low fiber ( < 10 g/day), high simple sugar (accounting for > 50% daily carbohydrate) diet for 7 days under the direction of a dietitian (Supplementary Fig. 5A).

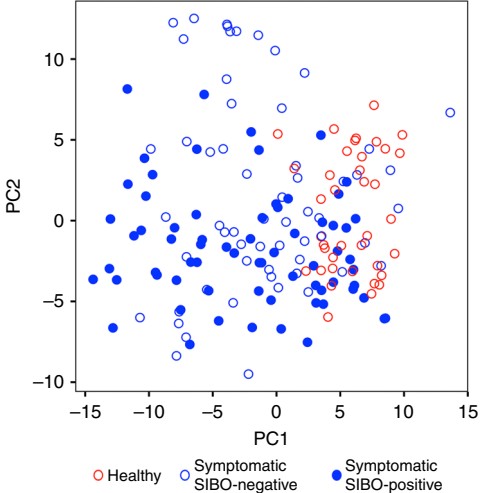

**Fig. 4** Quantitative small bowel culture does not reflect small bowel microbial composition. Distribution of microbial communities from symptomatic patients (blue) with and without SIBO based on Aitchison distance from healthy microbial communities (red). SIBO does not correlate with microbial community composition as summarized by dysbiosis classification ($p = 0.33$, Fisher's exact test). Open circles represent small bowel microbiomes from individuals who tested negative for SIBO; closed circles represent those tested positive for SIBO by aspirate culture

All food was provided to the subjects for the course of the study. We recorded the appearance of new symptoms following intervention and profiled microbial community diversity before and after intervention in duodenal aspirates and stool using 16 S rRNA marker-based sequencing. In addition, we measured changes in epithelial barrier function by fluorescein isothiocyanate (FITC)–dextran (4 kDa; measure of paracellular transport) flux across duodenal biopsies using an Ussing chamber.

As expected, the dietary intervention led to a decrease in fiber intake ($25 \pm 1.8$ g/day before; $9 \pm 0.18$ g/day after; $p < 0.0001$, paired $t$ test). All patients developed new symptoms with 80% developing GI symptoms during the dietary intervention and the symptoms resolved within a week of discontinuing the diet (Supplementary Table 3). Three subjects with baseline SIBO tested negative after the intervention, whereas five continued to have SIBO, and two subjects developed SIBO. SIBO resolution or new SIBO diagnosis following the intervention was not specifically associated with symptoms.

There was a significant correlation between the within-individual microbial community composition pre- and post intervention both in duodenal aspirate and stool by Procrustes analysis (Supplementary Fig. 5B, C). We found a significant inverse correlation between duodenal microbial phylogenetic diversity and duodenal permeability to 4 kDa FITC–dextran ($p = 0.02$, rho = −0.61, Spearman correlation; Fig. 6a, Supplementary Data 3), as well as decreased duodenal phylogenetic diversity in individuals who developed postprandial bloating ($p = 0.01$, Mann–Whitney $U$ test; Fig. 6b, Supplementary Data 3), or abdominal discomfort that was relieved by a bowel movement ($p = 0.07$, Mann–Whitney $U$ test; Fig. 6c, Supplementary Data 3).

To determine functional changes in the microbial communities following diet change, the metabolic content of duodenal aspirates and fecal samples collected before and after intervention was measured using $^1$H nuclear magnetic resonance (NMR) spectroscopy. Principal components analysis (PCA) was performed on these spectral profiles to identify biochemical variation associated with the intervention. Clustering was observed in the

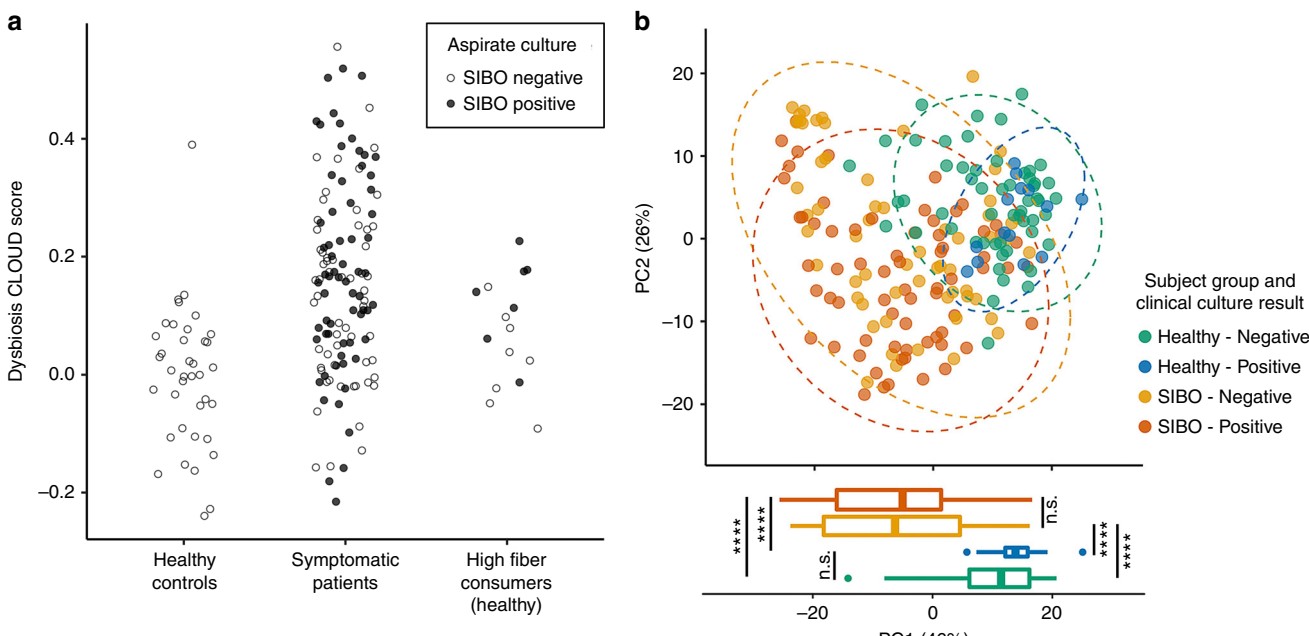

**Fig. 5** A subset of healthy individuals consuming high-fiber diet have SIBO. **a** DI and **b** distribution based on Aitchison distance of healthy controls without SIBO (green), symptomatic patients with (red) and without (orange) SIBO, and healthy individuals consuming a high fiber diet with (blue) and without (green) SIBO. ****, $q < 0.0001$; n.s., $q > 0.05$; pairwise $t$- test with FDR correction

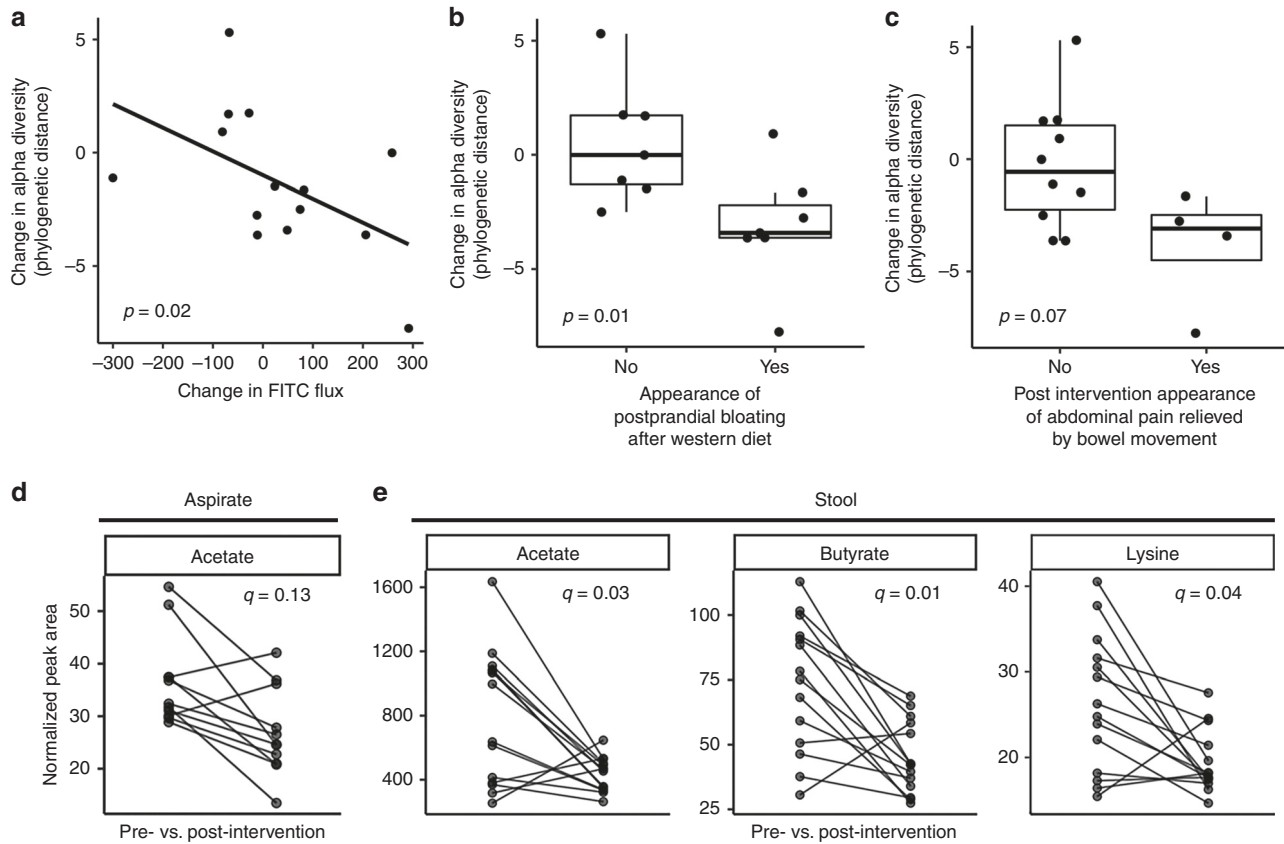

**Fig. 6** Diet change is associated with changes in host physiology, microbial diversity, and metabolites. **a** Spearman correlation of change in alpha diversity (PD whole tree) and change in duodenal permeability measured by FITC flux across duodenal biopsy in an Ussing chamber ($n = 14$, $p = 0.02$, rho = $-0.61$, Spearman correlation). Association of change in alpha diversity (PD whole tree) with **b** postprandial bloating ($n = 14$, $p = 0.01$, Mann–Whitney test) and **c** abdominal pain relieved by defecation ($n = 14$, $p = 0.07$, Mann–Whitney test). Tukey boxplots show the median with IQR and 1.5 IQR whiskers. Change in **d** acetate ($n = 11$, FDR $q = 0.1$, Wilcoxon signed rank test) in duodenal aspirates and **e** acetate ($n = 14$, FDR $q = 0.03$, Wilcoxon signed rank test), butyrate ($n = 14$, FDR $q = 0.01$, Wilcoxon signed rank test), and lysine ($n = 14$, FDR $q = 0.04$, Wilcoxon signed rank test) in stool measured before and after dietary intervention using $^1$H-NMR

scores plot (Supplementary Fig. 6A) from the PCA model comparing the pre- and post-intervention fecal profiles. This occurred along the first principal component (PC1) with the short chain fatty acids acetate, propionate and butyrate, the amino acids alanine and lysine, and succinate and glucose varying between the pre and post-intervention samples. The peak integrals for these metabolites were extracted from the NMR spectra (Supplementary Fig. 6C) of fecal samples and acetate, butyrate, and lysine were found to be significantly decreased in feces following the intervention (FDR $q < 0.05$, Wilcoxon signed rank test, Fig. 6e). Although no intervention-related clustering was apparent in the PCA model constructed on the duodenal aspirate profiles (Supplementary Fig. 6B), a decrease was observed in acetate in duodenal aspirates (FDR $q = 0.13$, Wilcoxon signed rank test, Fig. 6d). These changes are consistent with alterations in microbial energy processing and a loss of fermentable fiber sources from the diet. Changes in duodenal mucosal gene expression are described in supplementary information (Supplementary Fig. 8).

Separate correlation analyses were performed to identify statistical associations between the measured study variables before and after intervention. A number of significant associations were observed pre-intervention ($p < 0.05$, Spearman correlation, Supplementary Fig. 7A) including negative associations between fecal bacterial diversity (Chao1 and Shannon) and fecal acetate and propionate. In contrast, bacterial diversity in the duodenum (Shannon) was positively correlated with duodenal

acetate pre-intervention. Post intervention (Supplementary Fig. 7B), fecal bacterial diversity defined by the Simpson metric was negatively correlated with acetate in both the feces and aspirate samples while diversity measured by the Chao1 metric was positively correlated with duodenal glucose.

## Discussion

Here we show that SIBO based on quantitative duodenal aspirate culture usually reflects an overgrowth of anaerobes, does not correspond with patient symptoms, and can be a result of dietary preferences. Small intestinal microbial composition, however, is significantly altered in symptomatic patients when compared with healthy controls and the microbial composition does not correspond with SIBO. The differences in imputed microbial function in symptomatic patients and healthy controls suggest potential differences in dietary carbohydrate intake among the two groups. In our pilot dietary intervention study, we found switching from a high-fiber diet to low fiber, high simple-sugar diet triggered GI symptoms that may also be seen in FGIDs. There was an inverse correlation between small intestinal microbial diversity and small intestinal permeability and an association between decreased microbial diversity and appearance of specific GI symptoms following dietary intervention.

Our findings are consistent with previous reports that found the presence of SIBO did not correlate with symptoms such as diarrhea or abdominal distention, pain threshold of the colon or

increases in phasic contractions in colon in response to intra-luminal distension[25-27]. There is also a lack of data to support increased prevalence of SIBO in FGIDs like IBS[1]. The absence of correlation between SIBO and GI symptoms or microbial composition and the presence of SIBO in healthy individuals raises the question if SIBO is a primary driver of symptoms in FGIDs or represents an environmental influence. It will be important to make this distinction, as patients are currently treated with sub-therapeutic doses of non-targeted antibiotics on a recurrent basis based on the diagnosis of SIBO; these treatments have the potential for long-term collateral harm to the gut microbiome and are particularly concerning with the increasing rates of antibiotic resistance.

We found that small intestinal microbial composition was significantly altered in patients with GI symptoms, including significant decreases in members of genus *Prevotella* and enrichment of microbial ascorbate and aldarate metabolism. Ascorbic acid can act as an antioxidant and levels of total ascorbic acid in the blood decrease, whereas the oxidized form of ascorbic acid (dehydroascorbic acid) increases in cases of acute oxidative stress[28]. Prolonged oxidative stress has been associated with increased ascorbate catabolism[29,30]. There were several associated pathways that also showed higher expression in symptomatic patients, such as phosphotransferase systems required for microbial L-ascorbate import, gluconeogenesis, and the pentose phosphate pathway that have a role in ascorbate metabolism, and siderophore biosynthesis, which chelates iron. Together, these suggest a potential role for increased luminal oxidative stress in symptomatic patients, though this would need to be confirmed in a prospective study.

Host factors such as advanced age, PPI, antibiotic use, and prior GI surgery explain ~ 12% of variance in healthy and symptomatic patient microbial communities, suggesting there are other extrinsic or intrinsic factors at play. Diet has a significant influence on microbial composition and the enrichment of pathways linked to mono- and disaccharide metabolism in symptomatic patients supports this notion. Our pilot dietary intervention study showed that switching to a low fiber, high simple-sugar diet for a short period can trigger GI and systemic symptoms that improve upon resuming baseline diet. The correlation of decreased microbial diversity with increased duodenal permeability and appearance of symptoms suggests that this effect may be driven at least in part via gut microbiota. The deleterious effect of a low-fiber diet on microbial diversity and host function has been well described previously[31] and our findings support that notion. We found acetate and butyrate to be reduced with the low fiber, high simple-sugar diet. These microbial fermentation products are a key energy source for intestinal epithelial cells and play an important role in epithelial barrier integrity[32] and entero-pathogen exclusion[33].

We acknowledge that there are limitations that need to be addressed in future studies. The clinical metadata was obtained by retrospective chart review, limiting our ability to have complete follow-up and assess specific therapies instituted, the treatment response in all patients, and to obtain additional data such as diet history at the time of collection. The duodenal aspirates were collected using an aspiration catheter to reduce chances of contamination, however in spite of this; there is a chance for oro-pharyngeal contamination. The sample size of our pilot study was small, symptom questionnaire was not previously validated, and the duration of the intervention was short; but, our data provide a rationale for a controlled dietary intervention study in larger cohort with a longer intervention as well as post-intervention follow-up using standardized questionnaires. Despite the limitations, our findings provide a new avenue to pursue for patients with GI symptoms attributed to FGIDs including dietary

modification and potentially the use of targeted microbiota-directed therapies in patients with dysbiosis, which has the potential to improve our treatment outcomes.

In summary, our findings highlight the potential clinical benefit of characterizing the small intestinal microbiome. There are several mechanisms underlying the common GI symptoms such as diarrhea, bloating and abdominal pain, and gut microbiota is likely one such factor. Small intestinal microbial compositional analysis can help identify individuals in whom the microbiome is altered and may be contributing to symptoms as well as delineate individuals with healthy small intestinal microbiome where other factors may be at play. We also provide preliminary evidence supporting the interaction of diet with small intestinal microbiota in the development of GI symptoms commonly associated with FGIDs.

## Methods

**Ethical approval of human studies**. All human studies were approved by the Mayo Clinic IRB [test samples (16–006388 and 15–003235), healthy controls (14–002382 and 15–003603)], and we have complied with all relevant ethical regulations. The dietary intervention study was registered with ClinicalTrials.gov (NCT03266536). All subjects in the dietary study provided informed consent, whereas the small bowel aspirates were obtained from clinical microbiology laboratory under a consent waiver from the IRB for protocol 15–003235. Patient-derived biological materials cannot be shared based on protocol approved by institutional IRB.

**Collection of small intestinal aspirates**. Consecutive small bowel aspirate samples from symptomatic patients following diagnostic EGD were obtained directly from the microbiology laboratory from May 2016 through December 2016. Duodenal aspirate samples were collected during EGD following direct passage to the duodenum with minimal inflation in the stomach using a standard single lumen aspiration catheter passed through the suction port of the endoscope. Samples associated with patients who previously provided consent allowing for review of electronic medical record were included. A total of 143 aspirates from symptomatic patients were obtained over this time interval. After excluding samples due to lack of consent or low read depth on sequencing (< 1000) a total of 126 symptomatic patients was included. Duodenal aspirates obtained from 38 healthy volunteers participating in other research studies and collected in a similar manner were also obtained and processed similarly. Clinical metadata was obtained by retrospective review of electronic medical record including demographic information, BMI, clinical indication for SIBO testing, quantitative aerobic and anaerobic aspirate culture results, antibiotic course for treatment of SIBO, clinical response to antibiotics, need for repeat antibiotics, recent medications including antibiotics and PPI use, GI surgeries, comorbid conditions etc.

**Criteria for diagnosis of SIBO based on quantitative cultures**. Small bowel aspirates (10 μl) were streaked on RBAP (blood agar), REMB (gram-negative selection media), RBAPA (Pre-reduced blood agar) and incubated aerobically at 35 °C, 5% $CO_2$, RIMA (yeast selective media) and incubated at room temperature and on RCDC (anaerobic blood agar), RLKC (Laked blood Kanamycin Vancomycin—*Prevotella* selection), RPEA (phenol alcohol agar—gram-positive selective) and incubated anaerobically either in the clinical laboratory or research laboratory. Total bacteria (anaerobic plus aerobic counts) were reported as less than or greater than $10^5$ CFU/ml, and the diagnosis of SIBO was based on duodenal aspirate culture demonstrating ≥ $10^5$ CFU/mL bacterial growth (aerobic, anaerobic, or both) over 48 h[15–17].

**Comparison of cultures and 16S rRNA sequencing**. Bacterial DNA was extracted from aspirates and duodenal biopsies using phenol-chloroform, and from stool using a MoBio fecal DNA extraction kit, followed by 16 S rRNA amplification using Nextera library compatible primers flanking the V4 hypervariable region ([forward overhang] + 515 F: [TCGTCGGCAGCGTCAGATGTGTATAAGAGA CAG]GTGCCAGCMGCCGCGGTAA; and [reverse overhang] + 806 R: [GTCT CGTGGGCTCGGAGATGTGTATAAGAGACAG]GGACTACHVGGGTWTCT AAT) and prepared for sequencing using a dual-indexing protocol[34]. All samples were sequenced together in 2 × 300 paired-end mode on an Illumina MiSeq instrument using v3 reagents by the University of Minnesota Genomics Center.

Raw paired-end sequences were quality-filtered, adapter-trimmed, and stitched using the quality-control pipeline SHI7[35], with a trim threshold > 32 and mean quality score > 35. Preprocessed reads were analyzed by closed reference picking with the accelerated optimal gapped alignment engine BURST[36,37] in CAPITALIST mode against the NCBI RefSeq Targeted Loci Project bacterial and archaeal 16 S databases (https://www.ncbi.nlm.nih.gov/refseq/targetedloci/) at 97 percent identity. After filtering for samples with low read depth, we proceeded with analysis of 126 symptomatic patients and 38 healthy controls. Diversity analyses were done

using QIIME v1.9.1[38], and other data analyses, statistical tests, and visualizations were performed in R. To compare diversity of different GI tract sample types in the dietary intervention study, sequences were rarefied to the median depth of the site with the fewest reads, CLR (centered log-ratio)-transformed after multiplicative replacement, and analyzed with custom R scripts and the "vegan" package.

Differential abundance analysis between the healthy and symptomatic subjects was performed on normalized abundance data[39] at each taxonomic rank using a permutation test (1000 permutations) with t-statistic as the test statistic. Square-root transformation was applied to the normalized abundance data before testing. Taxa with prevalence < 10% or the maximum proportion < 0.2% were excluded from testing. FDR control (Benjamini–Hochberg procedure) was performed at each taxonomic rank to correct for multiple testing. The SI was generated using random forests classification based on the normalized OTU-level data ("randomForest" package in R). The index scale was defined as the out-of-bag probability of symptomatic patient classification based on the generated model. This method prevents overtraining by holding out the sample to be classified and predicting based on the remaining $n-1$ data set. Boruta feature[40] selection ("Boruta" package in R) was used to identify OTUs that contribute significantly to the classification. Associations with the seven demographic or clinical variables were tested using linear regression models and corrected for multiple testing using FDR control.

We generated the DI using the CLOUD method[24]. In brief, the Aitchison distance matrix of healthy samples was used as a reference cloud, and the distance of each healthy sample to the healthy cloud was calculated for every sample in the dataset. This distance calculation was then repeated for each symptomatic patient sample relative to the healthy cloud; distances greater than two standard deviations (SD) from the healthy distance mean are considered "dysbiotic"; those within 2 SD of the healthy distance mean are considered "healthy-like". This CLOUD dysbiosis score was then validated using an orthogonal approach based on the SI scores: we fit the healthy SI scores to a logit scale distribution (healthy control SI values are approximately normal on logit scale; $p = 0.41$, Shapiro–Wilk normality test). We again applied the 2 SD rule, classifying samples greater than 2 SD from the mean of the healthy distribution as "dysbiotic" and those within 2 SD as "healthy-like". To generate a DI here, we then re-classified the samples by random forest using the new group classifications, and used the OOB probability of "dysbiotic" classification as the index score. These two independent methods are in close agreement, as evaluated by the high degree of correlation between the samples' CLOUD dysbiosis scores and their probability-based index score ($r = 0.7$, $p = 2 \times 10^{-16}$, Pearson correlation). The CLOUD distance score was used in all subsequent analyses. As a comparison, we also grouped the data using spectral clustering, as implemented in the R package "kernlab" v0.9.27, using two cluster centers and default settings[41].

**Imputed functions from microbiota composition.** To predict microbial functions, preprocessed DNA reads were aligned to the GreenGenes 16 S rRNA database version 13_8 using BURST at 97% identity. The resulting OTU tables were used to generate predicted KEGG annotations using PICRUSt v1.1.3[42]. Linear discriminant analysis (LDA) effect size (LEfSe; Galaxy Version 1.0)[43] was then used to identify putative functions that differed significantly (LDA score > 2.0) between groups.

**Dietary intervention study in healthy volunteers.** This was a pilot prospective single-center dietary intervention study in healthy volunteers consuming a high-fiber diet at baseline. The study was registered with clinicaltrials.gov under NCT03266536. Eligible subjects were healthy adults ( ≥ 18 years) with baseline fiber intake ≥ 11 g/1000 calories/day; < 10% daily calories from added sugar; ≥ 5 servings of fruits and vegetables/day; and ≤ 13% daily calories from saturated fat based on completed food frequency questionnaire. Patients were excluded from the study if they did not meet the above diet requirements, had a known diagnosis of inflammatory bowel disease, microscopic colitis, celiac disease or other inflammatory conditions, presence of abdominal symptoms based on baseline questionnaire, oral antibiotic or probiotic use within the past 4 weeks, pregnancy or plans to become pregnant within the study time frame, or any other disease(s), condition(s), or habit(s) that would interfere with completion of study. Dietary instructions and food choices were discussed at the index, pre-intervention visit by a licensed dietician.

All subjects were screened in person or by phone and completed a food-frequency questionnaire to ensure they met inclusion and exclusion criteria. At the initial and follow-up visit, they underwent EGD with conscious sedation. Duodenal aspirates were obtained via a standard suction catheter passed through the suction port with minimal inflation in the stomach; an aliquot was submitted to clinical laboratory for testing for SIBO and the remaining was stored for microbiome analysis at − 80 °C. Eight duodenal biopsies were obtained for Ussing chamber studies, microbiome analysis and host RNA seq. At initial and follow-up visits, participants completed a symptom and demographic questionnaire and provided a stool sample. Symptoms assessed on questionnaires included: stool frequency, straining, incomplete evacuation, hard/lumpy stools, abdominal pain associated with bowel movements, diarrhea/loose-watery stools, bloating, swallowing difficulties, nausea/vomiting, heartburn, fatigue, and appetite. Answers were recorded in a binary (yes/no) fashion. For the intervention, all participants consumed a 7-day standardized diet with typical United States macronutrient

calorie distribution: 50% from carbohydrates, 35% from fats, and 15% from protein. The diet was low in fiber ( < 10 g/1000 calories/day) and high in simple sugar ( ≥ 50% of daily carbohydrates).

**Statistical power analysis.** We performed post-hoc power calculation for the comparison of the microbiome between healthy and symptomatic subjects, focusing on the power of differential abundance analysis, where we had much lower statistical power compared to alpha- and beta diversity analyses, due to multiple testing correction. We used the web-based microbiome power calculator to conduct power analysis (http://fedematt.shinyapps.io/shinyMB/), which was based on Monte Carlo simulations and Wilcoxon–Mann–Whitney test[44]. We used a false discovery rate of 5% to correct for multiple testing. Assuming that we are testing 65 genera with the abundance of 10 moderately abundant genera (abundance rank 6–15) decreasing by 50% in symptomatic patients (compared to 26 differential genera with a median decrease by 59% in the observed data), we had an average power of 75% to detect these 10 differential genera and a power of 100% to detect at least one significant genus at current sample size. Therefore, the study was reasonably powered to detect a moderate taxa difference when comparing the healthy and symptomatic subjects.

**Ex vivo epithelial barrier function.** Ussing chamber studies were performed to measure duodenal mucosal barrier function and secretory responses as has been previously described[45]. In brief, duodenal biopsies were mounted in 4 ml Ussing chambers (Physiologic Instruments, San Diego, CA) exposing 0.031 cm² area, within 45 min of collection. Chambers were filled with Krebs with 10 mM mannitol (mucosal side) and Krebs with 10 mM glucose (submucosal side). Baseline trans-epithelial resistance and short circuit current of each tissue was measured using a pair of Ag/AgCl electrodes with agar-salt bridges and a pair of current-giving platinum electrodes to maintain voltage clamp conditions. Paracellular flux across biopsies was measured using 4 kDa FITC–dextran administered on the mucosal side (1 mg/ml chamber concentration). Sampling was done from the submucosal side every 30 min for a total of 3 h and cumulative fluorescence was measured using a Synergy Multi-Mode Microplate Reader (BioTek, VT) and converted to concentration using standard curves. Cumulative flux at the end of 3 h was calculated for FITC–dextran. Spearman correlations (nonparametric) were used to test for a relationship between the microbiome diversity and FITC–dextran flux.

**RNA seq.** Biopsy samples were thawed on ice, homogenized, and centrifuged at 13,000 RPM, and the supernatant was transferred to a new tube. RNA was prepared using the RNeasy Mini Kit (QIAGEN, Hilden, Germany) and sequenced on an Illumina HiSeq 2500. Data analysis was done using Mayo Analysis Pipeline for RNA Sequencing (MAP-RSeq) with alignment to the human genome build (GRCh38.78)[46]. In brief, the MAP-RSeq pipeline preforms quality assessment of sequence reads from a FASTQ file, aligns the remaining reads with TopHat[47], and gene counts are aggregated with the Python HTSeq library[48]. Average total read depth was 57,052,595 (Standard deviation of $8.5 \times 10^6$) and on average 87.0% (Standard deviation of 2.0%) of the total reads mapped to gene models. Conditional quantile normalization was performed to account for gene length and GC content biases[49].

**¹H-NMR metabolomics sample preparation.** Aspirate samples were centrifuged at 4 °C at 12,000 × $g$ for 5 min. 162 μL of the supernatant was mixed with 18 uL of NMR buffer (1.5 M KH₂PO₄, 1 g/L of TSP and 0.13 g/L of NaN₃, Sigma-Aldrich) and transferred to a 3 mm NMR tube. Fecal samples were pre-weighed ( ~ 100 mg), randomized, and emptied into a screw-cap tube containing 50 mg of 1.0 mm Zirconia beads, to which 400 μL of ACN:H₂O (1:3) was added. The tube was placed in a Biospec bead beater for 30 s. The homogenized sample was then centrifuged for 20 min at 16,000 × $g$. The supernatant was carefully transferred into spin filter tubes and centrifuged for 30 min at 16,000 × $g$. 80 μL of the filtered fecal water was aliquoted into a 96-well plate, and 10 μL were used for quality control. Filtered fecal water in the 96-well plate was dried down under nitrogen flow before reconstituting with 540 μL of D₂O (Sigma-Aldrich) and 60 μL of the NMR buffer. The reconstituted fecal water and buffer mixture was transferred to 5 mm NMR experiment tubes.

**¹H-NMR experiment.** Metabolic profiles were measured on a Bruker 600 MHz spectrometer (Bruker Biospin, Rheinstetten, Germany) set at a constant temperature of 300 K for urine, aspirate and fecal samples and 310 K for plasma samples. A standard one-dimensional NOSEY experiment was performed for each urine, aspirate and fecal water sample and CPMG experiment for plasma samples. A total of 64 scans were acquired per sample into 64 K data points for urine, aspirates and plasma; 128 scans per sample were collected into 64 K data points for fecal water.

**¹H-NMR data pre-processing.** The spectra data were imported into MATLAB (Version 8.3.0.532 R2014a, Mathworks Inc, Natick, MA, USA). A series of in-house developed scripts was used for the following executions. Phasing, baseline corrections, and spectral calibration to TSP (0 ppm). The spectra were manually aligned. In order to account for the difference in sample concentration,

probabilistic quotient normalization was applied to the samples. In-house scripts were used to construct PCA and OPLS-DA models and integrate peaks of interest. Feature correlations and trends were analyzed in R.

**Reporting summary**. Further information on research design is available in the Nature Research Reporting Summary linked to this article.

## Data availability

All 16 S rRNA gene amplicon data and associated metadata that support the findings of this study are available through the European Nucleotide Archive with accession codes PRJEB31438 and PRJEB31439. Metabolomics data available through the EMBL-EBI MetaboLights database with the accession code MTBLS876, RNAseq data are available at GEO database under accession number GSE128189.

## Code availability

Previously published software packages and versions used to analyze microbiome, metabolomics, and RNAseq data are cited in the methods above. R code used for bioinformatic processing of the microbiome data and construction of the symptomatic and dysbiosis indices is available on GitHub (https://github.com/RRShieldsCutler/small_bowel_dysbiosis/).

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

## Acknowledgements
This work was made possible by funding from NIH DK111850 (PCK), DK114007 (PCK), DK115950 (MC), DK103911 (MG), and the Center for Individualized Medicine and Department of Medicine, Mayo Clinic, Rochester, MN (PCK).

## Author contributions
G.B.S., R.R.S.-C., D.K., J.L.S. and P.C.K. designed the experiments and the overall data analysis. G.B.S., R.R.S.-C., D.K. and P.C.K. wrote the manuscript with input from co-authors. G.B.S., M.G. and P.C.K. performed study procedures. H.R.L. recruited and followed all patients for the study. K.J.T. and K.K.K. performed RNAseq analysis. Y.Y. and J.R.S. performed metabolomics and data analysis. R.R.S.-C., J.C., V.L.H., J.J.F., G.B. and D.K. performed the microbiome bioinformatics analysis. J.M.C., E.J.B., Y.B., J.C.B., S.A.P., R.P., A.N.S. and M.G. performed microbiology and physiology experiments. G.F., M.G., M.C. and P.C.K. reviewed clinical data. All authors provided critical input in writing the manuscript.

## Additional information

**Competing interests:** Robin Patel: Dr. Patel reports grants from CD Diagnostics, BioFire, Curetis, Merck, Hutchison Biofilm Medical Solutions, Accelerate Diagnostics, Allergan, and The Medicines Company. Dr. Patel is or has been a consultant to Curetis, Specific Technologies, Selux Dx, GenMark Diagnostics, PathoQuest, Heraeus Medical and Genentech; monies are paid to Mayo Clinic. In addition, Dr. Patel has a patent on *Bordetella pertussis/parapertussis* PCR issued, a patent on a device/method for sonication with royalties paid by Samsung to Mayo Clinic, and a patent on an anti-biofilm substance issued. Dr. Patel receives travel reimbursement from ASM and IDSA and an editor's stipend from ASM and IDSA, and honoraria from the NBME, Up-to-Date and the Infectious Diseases Board Review Course. Michael Camilleri: Research support from Allergan to study effects of eluxadoline in IBS-diarrhea and bile acid malabsorption. Dan

Knights: DK serves as CEO and holds equity in CoreBiome, a company involved in the commercialization of microbiome analysis. The University of Minnesota also has financial interests in CoreBiome under the terms of a license agreement with CoreBiome. These interests have been reviewed and managed by the University of Minnesota in accordance with its Conflict-of-Interest policies. Purna Kashyap: Advisory board uBiome, ad hoc advisory board Salix Pharmaceuticals. The remaining authors declare no competing interests.

