## [Peer Review File · Nature Communications]

Reviewers' comments:

Reviewer #1 (Remarks to the Author):

This is an important study that concluded that small intestinal bacterial overgrowth (SIBO) based on duodenal aspirate culture reflects an overgrowth of anaerobes, does not correspond with patient symptoms, and may be a result of dietary preferences. Small intestinal microbial composition, on the other hand, is significantly altered in symptomatic patients and does not correspond with aspirate culture results. The authors also found that switching from a high fiber diet to a low fiber, high simple sugar diet triggered FGID-related symptoms and decreased small-intestinal microbial diversity and small-intestinal permeability. However, the study has some methodological issues.

1. "the lack of randomized controlled trials assessing the efficacy of specific antibiotics for SIBO" – This is not correct. There is a recent randomized controlled trial on this issue (Ghoshal UC et. al. Eur J Gastroenterol Hepatol. 2016 Mar; 28(3):281-9)

2. Phenotypes of the patients have not been presented adequately. What were the clinical diagnoses of the patients? Which criteria were used to arrive at the diagnosis? This is especially important as patients with short bowel syndrome, blind loop syndrome, post-surgical patients have different proximal gut microbiota than those with functional gastrointestinal disorders such as irritable bowel syndrome (IBS). Whereas patients with blind loop syndrome and similar conditions anaerobic bacteria would predominate, in other patients including IBS, anaerobes are less common. Moreover, a 52% frequency of presence of SIBO is also too high compared to the figure reported from any study on IBS raising concern about patients' phenotypes, which poses a major limitation to derive meaningful conclusions

3. Which catheter/tube was used for duodenal aspiration as a single lumen tube passed through the biopsy channel of the endoscope might result in contamination of the aspirate with the oropharyngeal bacteria

4. It is important to present data on which symptoms were correlated with which microbiota? For example, methanogens are associated with constipation as compared to the other bacterial which are associated with diarrhea and bloating. This issue also raises concern about paying more attention to the clinical data.

5. PPI, older age and GI surgery are well-known to cause dysbiosis including SIBO and are not really new observations though these do support the existing literature

6. The hypothesis on diet and microbiota is concerning as data dietary assessment might be confounded by several factors. Authors did acknowledge this major limitation by stating "Patients' dietary histories were not available as these samples were obtained directly from the clinical microbiology laboratory"

7. Authors may like to provide the statistical power of the sample size for the multiple aims that this study evaluated

We would like to thank the editor and the reviewers for their constructive criticism. We have conducted additional analyses to address the concerns raised by the reviewers and incorporated it into our manuscript. We feel this has significantly improved our manuscript.

Reviewer #1 (Remarks to the Author):

This is an important study that concluded that small intestinal bacterial overgrowth (SIBO) based on duodenal aspirate culture reflects an overgrowth of anaerobes, does not correspond with patient symptoms, and may be a result of dietary preferences. Small intestinal microbial composition, on the other hand, is significantly altered in symptomatic patients and does not correspond with aspirate culture results. The authors also found that switching from a high fiber diet to a low fiber, high simple sugar diet triggered FGID-related symptoms and decreased small-intestinal microbial diversity and small-intestinal permeability. However, the study has some methodological issues.

1. *“the lack of randomized controlled trials assessing the efficacy of specific antibiotics for SIBO” – This is not correct. There is a recent randomized controlled trial on this issue (Ghoshal UC et. al. Eur J Gastroenterol Hepatol. 2016 Mar;28(3):281-9)*

- Thank you for this comment and for highlighting one of the few RCTs assessing efficacy of antibiotics compared to placebo in SIBO. We have changed the wording in the sentence and cited the study to better reflect the state of the current literature.

2. *Phenotypes of the patients have not been presented adequately.*

- Thank you for this comment – we agree that the clinical data do require further clarification.

What were the clinical diagnoses of the patients? Which criteria were used to arrive at the diagnosis?

- As this was a retrospective review of prospectively collected samples we were limited by information available in the medical chart. We did an extensive review of the information available and during chart review, any associated clinical diagnoses for the patients were noted. We found associated clinic diagnoses clearly documented only in a minority of patients, and included celiac disease (14 patients), microscopic colitis (6 patients), ulcerative colitis (5 patients), and pancreatic insufficiency (5 patients). These diagnoses were not necessarily the etiology of the symptoms as patients were undergoing EGD to determine if SIBO was contributing to their current symptoms. However we agree that underlying

medical conditions may affect the microbiome. We conducted all analysis in non-targeted way, in order to prevent systematic bias. It is unlikely that these associated clinical conditions are driving microbiome differences. We did determine host factors that may contribute to variation in the microbiome, but given that an accompanying clinical diagnosis was present only in a minority of patients, it is difficult to ascertain their relative contribution. Nevertheless, we agree that it is important to include the diagnoses and this has now been included in the results section (page 5).

This is especially important as patients with short bowel syndrome, blind loop syndrome, post-surgical patients have different proximal gut microbiota than those with functional gastrointestinal disorders such as irritable bowel syndrome (IBS). Whereas patients with blind loop syndrome and similar conditions anaerobic bacteria would predominate, in other patients including IBS, anaerobes are less common.

- GI surgical procedures were abstracted on chart review and we have now added them to the results section (page 5). None of the patients were at risk for blind loop syndrome or short bowel syndrome based on the GI surgical history.

Moreover, a 52% frequency of presence of SIBO is also too high compared to the figure reported from any study on IBS raising concern about patients' phenotypes, which poses a major limitation to derive meaningful conclusions

- The true prevalence of SIBO remains unknown for a variety of reasons (absence of diagnostic clarity being a major issue). A recent review (Ghoshal UC et al. *Gut Liver*. 2017 Mar; 11(2):196 – 208) provided a summary table assessing the frequency of SIBO in IBS patients from a number of studies. This showed that the frequency ranges from 4% -78% depending on the diagnostic method implemented. Based on this, our value of 52% would not be considered too high and it is also similar to what our clinical microbiology laboratory has observed over the past years. It is important to clarify that these patients were not all diagnosed with IBS but rather were undergoing work up for symptoms commonly associated IBS.

3. Which catheter/tube was used for duodenal aspiration as a single lumen tube passed through the biopsy channel of the endoscope might result in contamination of the aspirate with the oropharyngeal bacteria

- Thank you for this comment. A sterile, single lumen catheter was indeed used and passed through the biopsy channel of the endoscope. Our current clinical practice in patients where small bowel aspirates need to be collected, is to advance the endoscope to duodenum without introducing significant air or suctioning in the stomach. We also mimicked this in our research-only patients; no suction of

contents was performed until the scope was positioned in the duodenum. The catheter was then passed to collect the small bowel aspirate. We acknowledge that in spite of this, there is a risk of oropharyngeal bacterial contamination with the method used for collecting aspirates, however this should not introduce a systematic bias as the methods employed were the same for all patients. We have now included this important limitation in the Discussion (page 13). We have also included the specifics of the catheter lumen in the Methods (page 14)

4. It is important to present data on which symptoms were correlated with which microbiota? For example, methanogens are associated with constipation as compared to the other bacterial which are associated with diarrhea and bloating. This issue also raises concern about paying more attention to the clinical data.

- Thank you for this comment. Upon chart review, the indications for SB aspirate testing did not include constipation, which would be an atypical manifestation of SIBO. As suggested by the reviewer we have now completed alpha-diversity, beta-diversity and differential abundance analyses with respect to individual symptoms within the symptomatic patients. We focused on the three major symptoms with numbers of patients greater than 10 (diarrhea [n = 51], abdominal pain [n=39] and bloating [n=19]). For alpha- and beta-diversity analyses, except abdominal pain, where we had marginally significant beta-diversity association (UniFrac $p = 0.048$, PERMANOVA), we did not observe other significant associations for individual symptoms. For differential abundance analyses, we could not identify differentially abundant taxa with a false discovery rate less than 10%. This is not entirely surprising as we are likely underpowered to detect a specific microbial signature for each symptom, given the smaller sample size within each symptoms group. There is definitely value in correlating individual symptoms with microbiome signature but these symptoms rarely occur in isolation and most patients with functional gastrointestinal disorders present with a constellation of symptoms and hence we included symptomatic patients as one group.

5. PPI, older age and GI surgery are well-known to cause dysbiosis including SIBO and are not really new observations though these do support the existing literature

- Thank you for this comment. We agree with the reviewer that association with these particular host factors is not novel, but at the same time it was important to include this to highlight that our cohort was similar to previously described literature. The novel aspect of the study indeed was the characterization of the microbiome and investigating potential role of diet.

6. The hypothesis on diet and microbiota is concerning as data dietary assessment might be confounded by several factors. Authors did acknowledge this major limitation by

stating “Patients’ dietary histories were not available as these samples were obtained directly from the clinical microbiology laboratory”

- Thank you for this comment. We agree that this is a limitation which could not be overcome and hence outlined in our discussion. We however tried to overcome this experimentally by conducting our prospective, pilot study, which now provides justification for a larger, controlled study to determine the role of diet in modulating small intestinal microbiome and GI symptoms.

7. Authors may like to provide the statistical power of the sample size for the multiple aims that this study evaluated

- The data on small intestinal microbiome is scarce which makes it difficult to estimate the effect size. Hence we did post-hoc power calculation for the comparison of the small intestinal microbial composition between healthy and symptomatic subjects. We focused on the power of differential abundance analysis, where we had much lower statistical power compared to alpha- and beta-diversity analyses, due to multiple hypothesis testing corrections. We used the web-based microbiome power calculator to conduct power analysis (<http://fedematt.shinyapps.io/shinyMB/>), which was based on Monte Carlo simulations and Wilcoxon-Mann-Whitney (WMW) test (Mattiello, Federico, et al. "A web application for sample size and power calculation in case-control microbiome studies." *Bioinformatics* 32.13 (2016): 2038-2040). We used a false discovery rate of 5% to correct for multiple testing. Assuming that we are testing 65 genera with the abundance of 10 moderately abundant genera (abundance rank 6-15) decreasing by 50% in symptomatic patients (cmp. 26 differential genera with a median decrease by 59% in the observed data), we had an average power of 75% to detect these 10 differential genera and a power of 100% to detect at least one significant genus at current sample size. Therefore, the study was reasonably powered to detect a moderate taxa difference when comparing the healthy and symptomatic subjects. For the pilot intervention study, we used a sample size similar to other microbiome based pilot studies (*Nature*. 2014 Jan 23;505(7484):559-63).

Reviewer #2 (Remarks to the Author):

Saffouri et al present study on characterization of small intestinal microbiome (SIM) in 126 patients with symptomatic gastrointestinal symptoms as well as 38 healthy controls. They indicate that SIM is notably altered in symptomatic patients and that better understanding of SIM may allow targeted antibacterial or diet based approach to gastrointestinal disorders (GD). Overall ,this is a well written paper and the findings are

encouraging and novel. In particular, observation that a low fiber diet triggers FGID-related symptoms and decreased small-intestinal microbial diversity is interesting. Still, even with intervention the results arise from a small study(only 15 subjects!!), and authors shall exercise caution with their conclusions as the fundamental question on causal role of the small intestinal microbiome is not answered in the study.

We would like to thank the reviewer for providing a critical analysis and we agree that the strength of our study is in the novelty of our findings.

Furthermore, diet - (microbial produced) metabolite interaction is not well described/analysed and shall be investigated in more details.

- We thank the reviewer for this comment. We appreciate that this depth of analysis was not explained in the manuscript and have updated the text to inform the reader how these metabolites were selected from the complex hyper-dimensional data. The metabolic phenotypes of these patients were investigated in detail pre- and post-intervention. Multivariate statistical techniques, such as principal components analysis, were performed to compare the complete fecal and aspirate metabolic profiles between pre- and post-intervention and have now been added as supplementary figures. These models identified a number of metabolites to differ in the fecal samples. For clarity, these metabolites were extracted (peak integration), analyzed with univariate statistics and presented in the main text (Fig 6E). We have provided this additional clarification in the text as below.

*“To determine functional changes in the microbial communities following diet change, the metabolic content of duodenal aspirates and fecal samples collected before and after intervention was measured using ¹H nuclear magnetic resonance (NMR) spectroscopy. Principal components analysis (PCA) was performed on these spectral profiles to identify biochemical variation associated with the intervention. Clustering was observed in the scores plot from the PCA model comparing the pre- and post-intervention fecal profiles (**Supplementary Fig. S6A**). This occurred along the first principal component (PC1) with the short chain fatty acids, acetate, propionate and butyrate, the amino acids, alanine and lysine, and succinate and glucose varying between the pre and post-intervention samples. The peak integrals for these metabolites were extracted from the NMR spectra (**Supplementary Fig. S6C**) of fecal samples and acetate, butyrate and lysine were found to be significantly decreased in feces following the intervention (FDR $q < 0.05$, **Fig. 6E**). Although no intervention-related clustering was apparent in the PCA model constructed on the duodenal aspirate profiles (**Supplementary Fig. S6B**), a decrease was observed in acetate in duodenal aspirates (FDR $q = 0.13$, **Fig. 6D**). These changes are consistent with alterations in microbial energy processing and a loss of fermentable fiber sources from the*

diet. Changes in corresponding duodenal mucosal gene expression are described in supplementary results (Supplementary Fig. S8)."

We also conducted additional correlation analysis to identify associations between microbiome characteristics and microbial metabolites.

"Separate correlation analyses were performed to identify statistical associations between the measured study variables before and after intervention. A number of significant associations ($p < 0.05$) were observed pre-intervention (Supplementary Fig. S7A) including negative associations between fecal bacterial diversity (Chao1 and Shannon) and fecal acetate and propionate. In contrast, bacterial diversity in the duodenum (Shannon) was positively correlated with duodenal acetate pre-intervention. Post intervention, fecal bacterial diversity defined by the Simpson metric was negatively correlated with acetate in both the feces and aspirate samples while diversity measured by the Chao1 metric was positively correlated with duodenal glucose (Supplementary Fig. S7B)."

Authors do state (line 186-187) they do not have sufficient evidence to support the role of diet-related changes in small intestinal microbiome in causing gastrointestinal symptoms. However beyond increasing the sample size of the intervention study, it will be tough provide better insights on microbe-metabolite interaction, and possibly causal role of SIM.

- We agree with the reviewers that this was a pilot study and hence the findings would need to be further validated in a large cohort. We chose the sample size based on prior dietary intervention studies investigating the microbiome, given the lack of prior data investigating dietary intervention and small intestinal microbiome. While our study sample size was small we did find interesting correlations between diet related changes in gut microbial diversity and metabolites and gastrointestinal symptoms, which justifies future controlled studies to validate these findings.

Below couple of technical suggestions that need to be addressed in order to publish the current paper

*1. Important aspect with respect to microbial data analysis in all of the reported experiments - OTU data was obtained using QIIME v1.9.1. as described in methods section. I strongly advice against using QIIME 1** due to known technical problems. For example see <https://peerj.com/articles/3889/> as even the developers of QIIME on their website recommend to switch to QIIME 2. This means underlying OTU data table used in all of the experiment could be unreliable. I would like to see the analyses being redone using Data2, usearch, mothur or qiime2 with data2 or deblur.*

- We thank the reviewer for this insight and indeed agree that QIIME 2 includes vast updates and improvements including those tools mentioned. We acknowledge that it was not sufficiently clear in the manuscript that our taxonomic profiling and OTU data, as described more thoroughly in the supplemental methods, was not performed with QIIME 1. We have improved the clarity of this material in the main text. We used the recently published DNA read quality processing pipeline SHI7 (Al-Ghalith et al. 2018 *mSystems*) and the exhaustive optimal gapped alignment engine BURST (Hillmann et al. 2018 *mSystems*; Vangay et al. 2018 *Cell*), querying a 16S rRNA database assembled using the NCBI Target Loci Project. QIIME 1 scripts were used solely to calculate certain diversity metrics, the results of which would not be impacted by upgrades in QIIME 2.

2. To determine the primary microbial determinants responsible for the difference in small intestinal microbial composition authors have used Random Forest classification on the OTU table. Reported AUC score is 0.896 - which is relatively high and may indicate overfitting of the model. In particular more care has to be taken with cross validation and leave-one-out estimate shall be considered. Also when training the model it is important to use stratified cross validation e.g. ensure that then number symptomatic patients and healthy controls is proportionally split among the folds. Because there is significant imbalance in group size [symptomatic patients (n=126) and healthy controls (n=38)] AUC may lead to over-optimistic estimates and PR curves might be more suitable <http://pages.cs.wisc.edu/~jdavis/davisgoadrichcamera2.pdf>

- We would like to thank the reviewer for the helpful comment as well as pointing us to an alternative to ROC curve when the group size is unbalanced. The OTU-based random forest (RF) was trained based on stratified bootstrap sample to address the problem of unbalanced groups. In the evaluation of the prediction error, we used out-of-bag (OOB) errors, which were based on the observations that were not part of the bootstrap sample used to train the RF model. The OOB error was shown to be a good estimator of the generalization error without the need for a set aside test set (Breiman, Leo. "Random forests." *Machine learning* 45.1 (2001): 5-32). A recent empirical study even found OOB error overestimated the generalization error (Janitza, Silke, and Roman Hornung. "On the overestimation of random forest's out-of-bag error." *PloS one* 13.8 (2018): e0201904). Thus, we decided to use the OOB error as a conservative measure to evaluate the prediction error. Since the sample size is not large, there is significant estimation uncertainty of the AUC. We have now provided the 95% confidence interval of AUC (95% CI: 0.844-0.949 (DeLong)) in the manuscript, together with the point estimate. To further alleviate the reviewer's concern about overfitting, we performed leave-one-out evaluation of prediction error. We again achieved a high AUC of 0.894 (95% CI: 0.841-0.947 (DeLong)). For the PR curve, we agree that it is more informative than the ROC curve when the number of negative examples (healthy)

greatly exceeds the number of positives examples (symptomatic) (Davis, Jesse, and Goadrich, Mark. "The relationship between Precision-Recall and ROC curves." Proceedings of the 23rd international conference on Machine learning. ACM, 2006). However, in our case, it is the other way around. We thus chose not to use the PR curve in our case.

3. *Characterisation of the dysbiosis via CLOUD method - is unclear and not very convincing. Why not to use standard clustering techniques such as spectral clustering* http://www.kyb.mpg.de/fileadmin/user_upload/files/publications/attachments/Luxburg07_tutorial_4488%5b0%5d.pdf

- We thank the reviewer for this comment and the suggestion to use spectral clustering. The CLOUD method has been recently published (Montassier et al “CLOUD: a non-parametric detection test for microbiome outliers” *Microbiome* (2018) <https://doi.org/10.1186/s40168-018-0514-4>) as a means to detect outlier communities based on distance to a control group, and is specifically developed for microbiome datasets. We did not observe clear clusters in the symptomatic patient microbiomes, but rather a gradient that included the healthy microbiome space. Since the CLOUD method is not clustering based, but uses a non-parametric neighborhood approach to identify significant deviation from a group (i.e. healthy controls) based on the multidimensional ecological distances of the communities, we chose to use this to distinguish patients whose microbiome falls outside the healthy control microbiome ‘cloud’ in ecological distance space. On the reviewer’s recommendation, we used spectral clustering on the same Aitchison distance matrix, using the implementation in the R package *kernlab*, with two cluster centers and default settings (Alexandros Karatzoglou, Alex Smola, Kurt Hornik, Achim Zeileis (2004). *kernlab - An S4 Package for Kernel Methods in R*. Journal of Statistical Software 11(9), 1-20. URL <http://www.jstatsoft.org/v11/i09/>). The result in fact shows reasonable agreement between the CLOUD method and the clusters from spectral clustering as shown in the contingency table below, where cluster 1 is analogous to the healthy-like communities and cluster 2 to the dysbiotic communities. We have added this information in Supplementary Table S3 and in the main text on page 7.

		CLOUD	
		dysbiotic	healthy-like
spectral clustering	cluster 1	9	119
	cluster 2	29	7

REVIEWERS' COMMENTS:

Reviewer #1 (Remarks to the Author):

The revision has been satisfactory

Reviewer #2 (Remarks to the Author):

The authors have answered all my questions satisfactorily